# Effect of Organic Assistant on the Performance of Ceria-Based Catalysts for the Selective Catalytic Reduction of NO with Ammonia

**Xing Huang †, Shining Li †, Wenge Qiu \*, Yun Chen, Jie Cheng, Yanming Sun, Guangmei Bai, Liyun Song, Guizhen Zhang and Hong He \***

Beijing Key Laboratory for Green Catalysis and Separation, Key Laboratory of Beijing on Regional Air Pollution Control, Department of Chemistry and Chemical Engineering, College of Environmental and Energy Engineering, Beijing University of Technology, Beijing 100124, China; tuhuangxingtc@126.com (X.H.); lsn9295@126.com (S.L.); chenyun053@126.com (Y.C.); chengjie7208@163.com (J.C.); yanming_sun0917@163.com (Y.S.); Baiguangmei@bjut.edu.cn (G.B.); songly@bjut.edu.cn (L.S.); zhangguizhen@bjut.edu.cn (G.Z.)

\* Correspondence: qiuwenge@bjut.edu.cn (W.Q.); hehong@bjut.edu.cn (H.H.)

† The first two authors contributed equally to this work and are joint first authors.

**Abstract:** In the present study, a series of $CeO_2/TiO_2$ catalysts were fabricated by dry ball milling method in the absence and presence of organic assistants, and their catalytic performances for the selective catalytic reduction (SCR) of NO by $NH_3$ were investigated. It was found that the addition of organic assistants in the ball milling process and the calcining ambience exerted a significant influence on the catalytic performances of $CeO_2/TiO_2$ catalysts. The nitrogen sorption isotherm measurement (BET), powder X-ray diffraction (XRD), Raman spectra, high-resolution transmission electron microscopy (HR-TEM), hydrogen temperature-programmed reduction ($H_2$-TPR), ammonia temperature-programmed desorption ($NH_3$-TPD), sulfur dioxide temperature-programmed desorption ($SO_2$-TPD), thermogravimetric analysis (TG), Fourier transform infrared (FT-IR) and X-ray photoelectron spectra (XPS) characterizations showed that the introduction of citric acid in the ball milling process could significantly change the decomposition process of the precursor mixture, which can lead to improved dispersion and reducibility of cerium species, surface acidity as well as the surface microstructure, all which were responsible for the high low temperature activity of CeTi-C-N in an $NH_3$-SCR reaction. In contrast, the addition of sucrose in the milling process showed an inhibitory effect on the catalytic performance of $CeO_2/TiO_2$ catalyst in an $NH_3$-SCR reaction, possibly due to the decrease of the crystallinity of the $TiO_2$ support and the carbon residue covering the active sites.

**Keywords:** selective catalytic reduction (SCR); ceria-based catalysts; organic assistant; ball milling

## 1. Introduction

Nitrogen oxides ($NO_x$) primarily emitted by combusting fossil fuels have been a major source of air pollution that has caused serious environmental problems, e.g. acid rain, urban smog and haze weather. Several methods have been used for $NO_x$ abatement. The optimization of combustion process is the prioritized way to minimize the $NO_x$ output [1], whereas highly efficient catalytic abatement technologies are increasingly required to satisfy increasingly stringent regulations. Selective catalytic reduction (SCR) of $NO_x$ with $NH_3$ is the method to most effectively reduce the emission of $NO_x$ from stationary and mobile sources [2–4]. Since the 1970s, $V_2O_5$-$WO_3/TiO_2$ and $V_2O_5$-$MoO_3/TiO_2$ have served as the main commercial SCR catalysts because of their high efficiency for $NO_x$ reduction and

low sensitivity towards $SO_2$ poisoning [5,6]. Nevertheless, these catalysts still suffer from the toxicity of $V_2O_5$ to the environment, inferior performance at low temperatures, and the high-temperature over-oxidation of $NH_3$ to $N_2O$. As a result, many transition metal oxides catalysts (e.g. $CeO_2$, $MnO_x$, and $Fe_2O_3$) have been developed [7–10], among which ceria is considered a promising active substrate because of its high oxygen storage capacity (OSC) and excellent redox properties. Different cerium-based catalysts, e.g. $CeO_2/TiO_2$ [11], $CeO_2$-$MnO_x$ [12], and $CeO_2$-$WO_3$ [13], were reported with high SCR activity and $N_2$ selectivity, whereas these catalysts showed gradually decreased activities in the presence of $SO_2$ and $H_2O$ owing to the blocking of the active sites [14–16]. Accordingly, a high resistance to $SO_2$ and $H_2O$ poisoning must be of concern for low-temperature SCR catalysts. It is known that optimization synthesis techniques are an important strategy to improve the performance of cerium-based catalysts. Homogeneous precipitation [17], sol-gel [18], co-precipitation [19] and flame-spray synthesis [20] have been all proven as efficient preparation methods to improve the properties of ceria-based catalysts.

Recently, mechanochemical synthesis has become more and more important and was widely used in the preparation of nano-materials and heterogeneous catalysts due to its advantages of solvent-free reactivity, high energy efficiency and environment friendly processes compared to the traditional solution-based methods [21,22]. In our previous work, we found that adding adipic acid during the milling process can improve the dispersion of cerium species on the support, the reducibility, the $Ce^{3+}/(Ce^{3+}+Ce^{4+})$ ratio and the surface structure of the cerium-based catalyst [23,24], which promoted $NH_3$-SCR reactions. It seems that the "ligand-assisted ball milling" may offer the discovery of new or improved reactivity and products. In order to explore whether the linear ligand can be expanded to other ligand molecules and general organic compounds, a traditional complexant (citric acid) and a common organic compound (sucrose) were chose as the additive. In the present study, a series of $CeO_2/TiO_2$ catalysts were prepared by dry ball milling in the presence of citric acid and sucrose, respectively, and their activities for the SCR reaction with $NH_3$ as the reducing agent were studied. All catalysts were characterized using different techniques.

## 2. Results and Discussion

### 2.1. Catalytic Activity

Figure 1 shows the catalytic performance of the $CeO_2/TiO_2$ catalysts for the SCR reaction of NO by $NH_3$. The CeTi-C-N catalyst exhibited the best low temperature activity for NO removal (Figure 1a), and the NO conversion was up to 84.6% at 180 °C, which was much higher than the data of CeTi-S-N (35.1%) at the same temperature. The CeTi-N sample that was prepared under a similar ball milling condition without the organic assistant exhibited a moderate low temperature activity and about 51.3% NO was converted at 180 °C. The results indicated that adding citric acid during the milling process had an obvious promotion on the low temperature activity of $CeO_2/TiO_2$ catalysts. In contrast, the addition of sucrose had an inhibitory effect. When the calcining condition changed from nitrogen atmosphere to air, similar effects of organic assistants on the low temperature activity were also observed, indicated by the distinction between the activities of CeTi-A, CeTi-C-A and CeTi-S-A at low temperature range (Figure 1a). In addition, it could be observed a significant distinction of NO conversion in the high temperature between the $CeO_2/TiO_2$ catalysts calcined under nitrogen and air atmosphere, respectively. The NO conversions over CeTi-A, CeTi-C-A and CeTi-S-A at 450 °C were 80.9%, 84.7%, 82.1% and 72.7%, respectively, which was much higher than that of CeTi-N (64.9%), CeTi-C-N (24.0%) and CeTi-S-N (18.1%) at the same temperature.

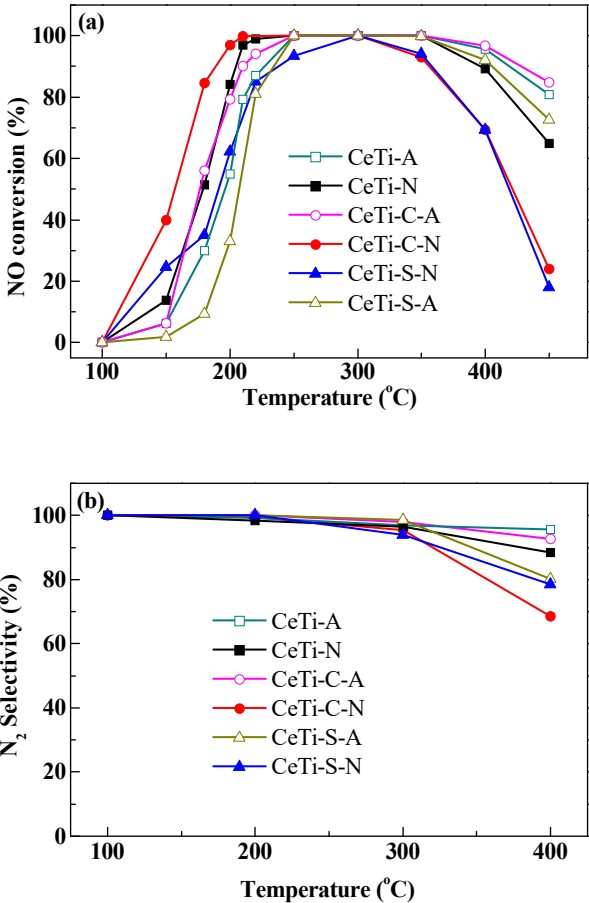

**Figure 1.** NO conversions (**a**) and $N_2$ selectivity (**b**) as a function of reaction temperature over the $CeO_2/TiO_2$ catalysts. Reaction conditions: $[O_2]$ = 6 vol%, $[NO]$ = $[NH_3]$ = 1000 ppm, balance $N_2$, total flow rate = 500 ml/min, GHSV = 30,000 h$^{-1}$.

The $N_2$ selectivity results were shown in Figure 1b. All the $CeO_2/TiO_2$ catalysts exhibited high $N_2$ selectivity in the low temperature range (< 300 °C). There almost no $N_2O$ was detected in this temperature range, but it is noteworthy that the $N_2$ selectivity decreased gradually as the temperature further increased (Figure 1b) due to side reactions of ammonia oxidation [25]. At 400 °C, CeTi-A exhibited the best $N_2$ selectivity, and CeTi-C-N exhibited the worst $N_2$ selectivity among the six catalysts. It could be concluded that the addition of organic assistants during the milling process and the calcining ambience exerted a significant influence on the catalytic activities of $CeO_2/TiO_2$ catalysts.

It was reported that the $H_2O$ and $SO_2$ exhibited a synergetic inhibition on NO reduction over $CeO_2/TiO_2$ at lower temperature [26]. The activity data of the $CeO_2/TiO_2$ catalysts in the presence of $H_2O$ and $SO_2$ showed that the NO conversions at low temperature (below 200 °C) over all catalysts decreased notably (Figure S1). However, the NO conversion over CeTi-C-N was still close to 70%, which was much higher than that of the other catalysts, implying that CeTi-C-N catalyst exhibited relatively better $H_2O$ and $SO_2$ resistance comparing to the other ones. It is noteworthy that the activity order of CeTi-S-A and CeTi-S-N were reversed when $SO_2$ and $H_2O$ were introduced into the simulated exhaust gases.

## 2.2. Structural and Textural Characterization of Various Catalysts

The porosity and specific surface area of the as-prepared $CeO_2/TiO_2$ catalysts were characterized by $N_2$ adsorption-desorption isotherms (Figure S2). It could be seen that all the catalyst samples exhibited type IV isotherms and typical H2 hysteresis loops, suggesting the mesoporous structure of the catalysts (Table 1) [27]. All the samples except CeTi-S-N displayed a capillary condensation

at high relative pressure $P/P_0$ (about 0.80), demonstrating a characteristic of inter-crystal porosity. The hysteresis loop of CeTi-S-N moved to low relative pressure $P/P_0$ between 0.45 and 0.8, indicating the sorption behavior of mesopores. But its average pore diameter (5.7 nm) was even lower than the others possibly due to the blocking of partial orifices by the carbon residue. The data showed that the specific surface areas and pore volumes of CeTi-C-A, CeTi-C-N, and CeTi-S-A were larger than those of CeTi-A and CeTi-N (Table 1), which revealed that the presence of organic assistants can lead to the variations of surface and pore structures of the $CeO_2/TiO_2$ catalysts. However the surface area order of the catalysts was not consistent with their $NH_3$-SCR activity at low temperature, which suggested that the surface area of these catalysts might be not the factor dominating the catalytic activity in the $NH_3$-SCR reaction.

**Table 1.** Textural and structural properties of the $CeO_2/TiO_2$ catalysts and the $TiO_2$ support.

| Samples | BET Surface Area ($m^2/g$) | Total Pore Volume ($cm^3/g$) | Average Pore Diameter (nm) |
|---|---|---|---|
| CeTi-A | 50 | 0.11 | 13.0 |
| CeTi-N | 44 | 0.11 | 12.9 |
| CeTi-C-A | 77 | 0.19 | 10.6 |
| CeTi-C-N | 77 | 0.17 | 9.1 |
| CeTi-S-A | 74 | 0.2 | 12.8 |
| CeTi-S-N | 108 | 0.13 | 5.7 |
| $TiO_2$ | 77 | 0.43 | 16.1 |

The X-ray diffraction (XRD) patterns of the $CeO_2/TiO_2$ catalysts prepared from different conditions are shown in Figure 2. For the samples of CeTi-A, CeTi-N, CeTi-C-A, CeTi-C-N and CeTi-S-A, the most peaks were attributed to anatase $TiO_2$, indicating that the $TiO_2$ support maintained its initial phase structure. In the meantime, a wide weak diffraction peak at 28.6° due to cubic $CeO_2$ was detected, showing that the $CeO_2$ on the support had a small crystallite size. Furthermore, the diffraction peaks of $CeO_2$ in CeTi-A and CeTi-C-A were relative sharper than that in CeTi-N and CeTi-C-N, respectively. For the CeTi-A and CeTi-C-A samples, a weak sharp peak at $2\theta = 26.6°$ was also detected, indicating the formation of a bit of rutile $TiO_2$ (No. 21-1276). However, the intensities of the diffraction peaks assigned to anatase $TiO_2$ in the CeTi-S-N sample were very weak, and the cubic $CeO_2$ phase was undetectable, showing the decrease in the degrees of crystallinity of $TiO_2$ and an amorphous cerium species on the $TiO_2$ surface were possibly due to the presence of carbon residue from the carbonization of sucrose. The mentioned results implied that the addition of organic assistants during the milling process and the calcining atmosphere had significant effects on the crystalline phase structure of $CeO_2$ and $TiO_2$.

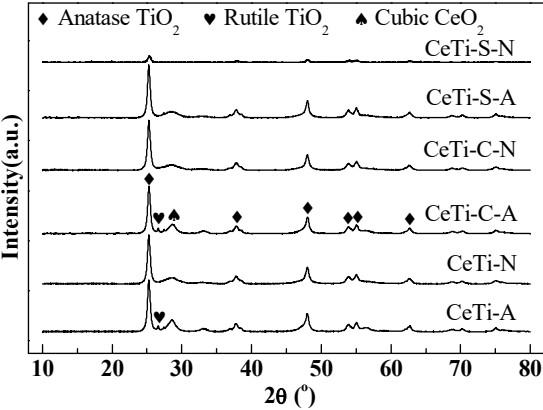

**Figure 2.** XRD patterns of the $CeO_2/TiO_2$ catalysts.

To further investigate the structure difference of the catalysts, the Raman spectroscopy was performed (Figure 3). Pure $CeO_2$ had only a prominent band at ca. 464 cm$^{-1}$ corresponding to the $F_{2g}$ vibration mode of cubic fluorite structure of $CeO_2$ [28]. However, this peak was not found in all the $CeO_2/TiO_2$ catalysts, implying that the ceria species in the catalysts were not Raman active [29]. In contrast, in the Raman spectra, 5 peaks were observed at 143($E_g$), 199($E_g$), 397($B_{1g}$), 518($A_{1g}$) and 639($E_g$) cm$^{-1}$ in $CeO_2/TiO_2$ catalysts, which all assigned to the typical characteristic of anatase crystalline [30]. In comparison with CeTi-C-A and CeTi-S-A, the main $E_g$ peak at 143 cm$^{-1}$ of CeTi-C-N and CeTi-S-N had an obvious blueshift and a relatively large half-peak width, indicating the decrease of $TiO_2$ particle size [30]. This phenomenon indicated the structure difference of the samples that were prepared under a different calcining atmosphere. In addition, it could be observed that the addition of organic assistants during the milling process has a significant effect on the intensity of Raman peaks. CeTi-C-A and CeTi-C-N had a strong $E_g$ peak at about 143 cm$^{-1}$. In contrast, the intensities of the Raman peaks of CeTi-S-A and CeTi-S-N were even weaker than CeTi-A and CeTi-N. In general, the larger the content of the symmetric vibration bonds, the stronger the Raman signals. According to this result, the strong interaction between citric acid and Ce species as well as $TiO_2$ supports might improve the formation of more symmetric surface structures, which did not exist in the presence of sucrose. These results were consistent with the XRD data and the catalytic activities. Two typical bands of carbon species at around 1367 cm$^{-1}$ (D band) and 1596 cm$^{-1}$ (G band) are observed in the Raman spectra (Figure 3b) of CeTi-C-N and CeTi-S-N as well, indicating the presence of carbon residue in the samples.

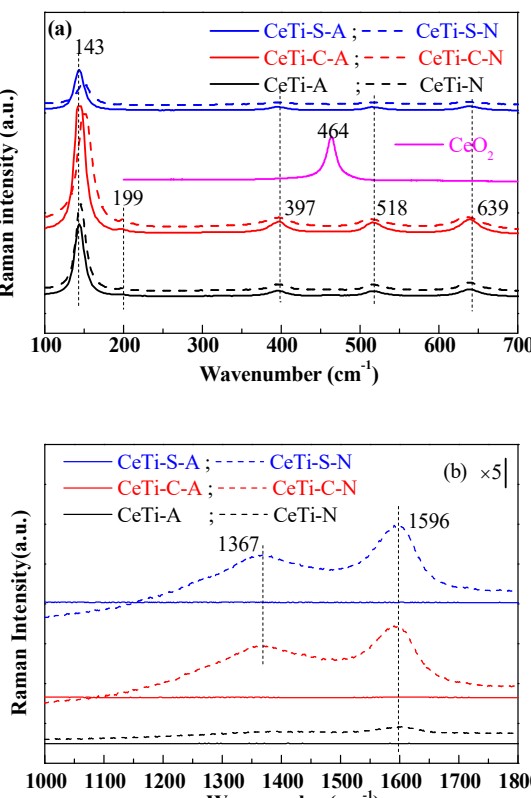

**Figure 3.** Raman spectra of series $CeO_2/TiO_2$ catalysts at wavelength coverage: 100–700cm$^{-1}$ (**a**) and wavelength coverage: 1000–1800 cm$^{-1}$ (**b**).

In order to determine the particle morphology and the existing form of carbon residue in part of the $CeO_2/TiO_2$ catalysts, high-resolution transmission electron microscopy (HR-TEM) characterizations were performed, and the images are given in Figure 4. The crystalline phase structures of $TiO_2$ and $CeO_2$ were determined by the interplanar distance measurement. Through careful searching, the lattice

fringes of both anatase TiO$_2$ (101) and cubic CeO$_2$ (111) crystal planes with interplanar spacing of 0.35 and 0.31 nm were well-defined in the sample of CeTi-A and CeTi-C-A. Furthermore, the fewer rutile TiO$_2$ (110) crystal planes with interplanar spacing of 0.33 nm were also detected. Differently, the mean particle diameter for CeTi-C-A was about 7–10 nm, which was smaller than that of CeTi-A (8–20 nm), showing that adding citric acid during the ball milling could lead to a higher dispersion of CeO$_2$ nanoparticles on the TiO$_2$ support compared with the process without organic assistant. For CeTi-C-N (Figure 4c), the anatase TiO$_2$ (101), and cubic CeO$_2$ (200) and (111) crystal planes with interplanar spacing of 0.35 nm, 0.26 nm and 0.31 nm, respectively, could be found, which was different from CeTi-S-N. In the later, there was only slightly anatase TiO$_2$ (101) and cubic CeO$_2$ (200) could be detected. This result was also consistent with XRD. In addition, the amorphous zone of carbon was significant on the surface of CeTi-C-N and CeTi-S-N. However, the amorphous carbon covered over half of the surface area of CeTi-S-N, and even curtained off the support and active component. On CeTi-C-N, only thin carbon membrane and a few pieces of free amorphous carbon that might dropped from the catalyst were observed. This evidence further confirmed the inhibition of carbon residue in the CeTi-S-N sample.

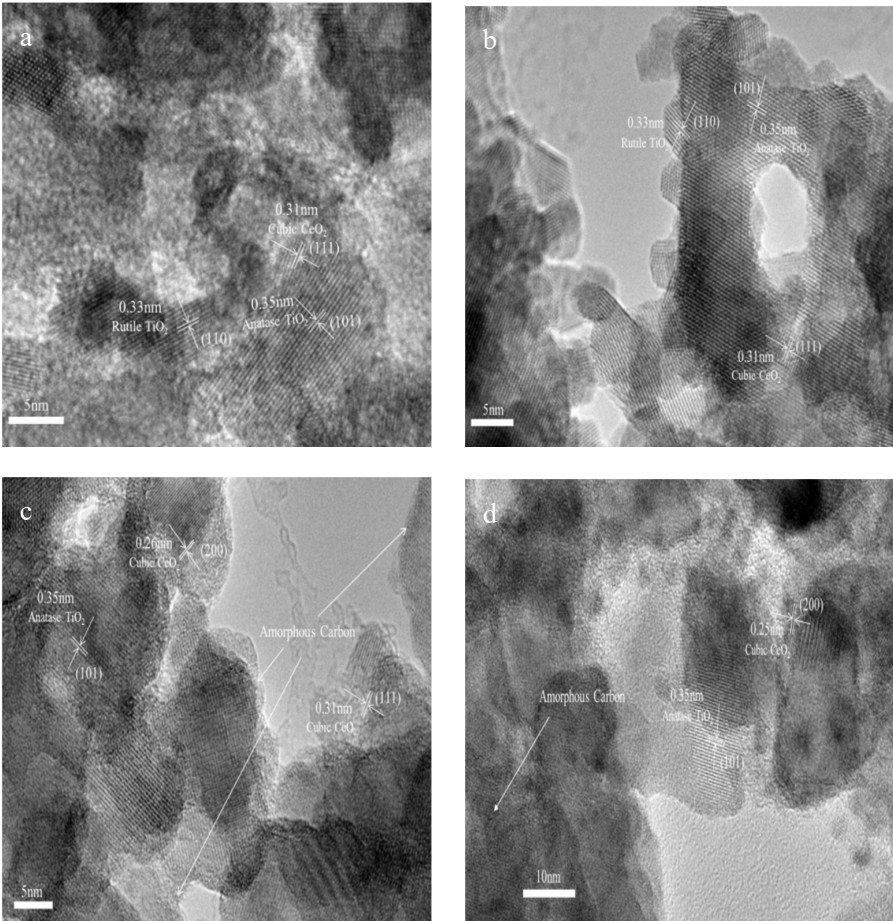

**Figure 4.** TEM images of (**a**) CeTi-A, (**b**) CeTi-C-A, (**c**) CeTi-C-N, and (**d**) CeTi-S-N catalysts.

### 2.3. Measurements of the Decomposition Process of Organic Assistants (TG and FT-IR)

The TG curves of the mixtures of Ce(NO$_3$)$_3$/TiO$_2$/citric-acid and Ce(NO$_3$)$_3$/TiO$_2$/sucrose, which were the precursors of CeTi-C-A or CeTi-C-N and CeTi-S-A or CeTi-S-N, respectively, are shown in Figure 5. The pyrolysis process of the grinded powder of Ce(NO$_3$)$_3$/TiO$_2$/citric-acid included three steps (Figure 5a). In the first step, the continuous mass loss of 9.7% up to about 140 °C was attributed to the removal of crystal water, which matched well with the calculated data of 9.3%. The

main mass loss of 16.7% in $N_2$ condition in the temperature range of 140–200 °C was assigned to the decomposition of nitric acid, which was a little bit higher than the calculated data of 12.4% due to the decomposition of partial citrate molecules in this step. In air condition, the mass loss was 20.4% in the same temperature range (step 2), which was much larger than the calculated data (12.4%), implying that more citrate molecules were decomposed in the presence of oxygen. After that, the rates of mass loss became slow in the temperature range of 200–250 °C due to the formation of coordination compounds between cerium ions and citric acid molecules, which started to decompose quickly at about 250 °C. The residual mass fractions in $N_2$ condition (59.6%) and that in air condition (52.8%) after heating to 500 °C were slightly smaller than the calculated values 64.7% and 55.2%, respectively. However, the TG curves of the mixtures of $Ce(NO_3)_3/TiO_2/sucrose$ were significantly different from that of the precursor of CeTi-C-A or CeTi-C-N, which could be divided into four steps (Figure 5b). The mass loss of 6.7% up to about 160 °C was assigned to the removal of crystal water, which matched well with the calculated value of 6.9%. It was known that sucrose could undergo intermolecular dehydration and aldol condensation in the temperature range of 150–250 °C, which leads to polymerization and carbonization of sucrose [31], so the main mass loss of 9.6% in the temperature range of 160–230 °C was attributed to the partial dehydration and carbonization of sucrose, which was much lower than the data (27.9%) corresponding to the total decomposition of sucrose. In our previous work, we found that cerium nitrate started to decompose at 230 °C [24]. The main mass loss (11.4%) in the third step between 230-260 °C could be assigned to the partial decomposition of cerium nitrate. This deduction was also confirmed by the TG data of the mixture of $Ce(NO_3)_3/TiO_2$ (Figure S3). After that, the rates of mass loss became slow in the high temperature range corresponding to the further decomposition of nitrate and residual organic matter. In the whole temperature range, the pyrolysis processes of the mixture of $Ce(NO_3)_3/TiO_2/sucrose$ in $N_2$ and air conditions, respectively, were similar, but the rate of mass loss in $N_2$ condition was a little bit slower than that in air condition, revealing the booster action of oxygen.

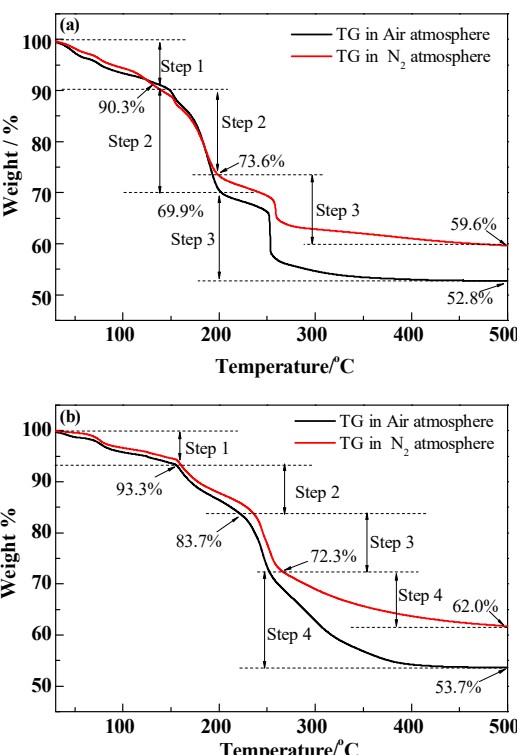

**Figure 5.** TG profiles of the mixtures of $Ce(NO_3)_3/TiO_2/citric$-acid (**a**) and $Ce(NO_3)_3/TiO_2/sucrose$ (**b**) under air or $N_2$ atmosphere, respectively.

The FT-IR spectra of the mixture of $Ce(NO_3)_3/TiO_2$/citric-acid treated at different temperatures for 2 h in air (Figure S4a) showed the coordination between cerium ion and citric acid molecule, indicated by the changes of the intensity of stretching vibration of carboxylic group and the appearance of new bands at 1628 and 1429 $cm^{-1}$. The band at 1310 $cm^{-1}$ corresponded to the asymmetric stretching vibration of nitrate ions. This band almost disappeared at 200 °C, revealing the decomposition of nitrate. For the $Ce(NO_3)_3/TiO_2$/sucrose system (Figure S4b), the FT-IR curves of the mixture even treated at 100 °C were similar to that of pure sucrose, implying that there was relatively weak interaction between cerium ion and sucrose. The TG and FT-IR data showed that the decomposition process of the mixture of $Ce(NO_3)_3/TiO_2$/citric-acid was quite different from that of the mixture of $Ce(NO_3)_3/TiO_2$/sucrose. The coordination between cerium ion and citric acid molecule can have a significant effect on the microstructure of cerium species that were recently formed [32].

### 2.4. Redox Properties, Surface Acidity and $SO_2$ Molecular Adsorption on Surface ($H_2$-TPR, $NH_3$-TPD and $SO_2$-TPD)

The reducibility of the catalysts was detected using $H_2$-TPR, which is correlated with the low-temperature activity of $NH_3$-SCR catalysts [33]. Figure 6 shows the $H_2$-TPR profiles of the various catalysts calcined in the atmosphere of $N_2$ and air. Since the reduction of titanium oxides is known to be harder than that of cerium below 700 °C, the peaks of the various catalysts could be attributed to the reduction of $CeO_2$. It could be seen that all of the three samples calcined in air showed two reduction peaks at 538/451/480 °C and 681/605/607 °C (Figure 6b), respectively, which could be ascribed to the reduction of surface oxygen of stoichiometric cerium ($Ce^{4+}$-O-$Ce^{4+}$) and bulk $CeO_2$, respectively [34,35]. The two reduction peaks of CeTi-C-A and CeTi-S-A significantly shifted towards low temperature, suggesting that the addition of organic assistants in the process of ball-milling resulted in the $CeO_2/TiO_2$ catalysts becoming more easy to reduce and implying the enhancement of redox activity of the corresponding catalysts due to a higher $CeO_2$ dispersion. For the catalysts calcined in $N_2$ atmosphere (Figure 6a), CeTi-N had two broad reduction regions at 70–250 °C and 450–750 °C, which corresponded to the reduction of the adsorbed oxygen on the cerium surface and lattice oxygen in bulk $CeO_2$, respectively. Moreover, the total amount of $H_2$ consumption was relatively low (Table S1), which reveals that the relative amount of reducible cerium species in CeTi-N was low. For CeTi-S-N, there was only one strong peak at 464 °C, indicating the easily reducibility of the $CeO_2$ species on the surface of $TiO_2$ support. CeTi-C-N showed a main reduction peak at 445 °C and a shoulder peak at 489 °C, and the total reducible amount of $CeO_2$ was similar to CeTi-S-N (Table S1). It could be seen that the total $H_2$ consumptions of CeTi-A, CeTi-C-A and CeTi-S-A were higher than that of CeTi-N, CeTi-C-N and CeTi-S-N, respectively, showing that calcining treatment at $N_2$ atmosphere could adjust the content of surface oxygen species and the ability of oxygen release that also reflected the reducibility of catalysts. The relatively low reduction temperatures of CeTi-C-A and CeTi-C-N were consistent with their outstanding low temperature $NH_3$-SCR performance. But the good reducibility of CeTi-S-A and CeTi-S-N was conflicting with their bad low temperature activity. This could be attributed to the significant variation of the crystallinity of $TiO_2$ and the carbon residue covering the part active sites on the catalyst surface.

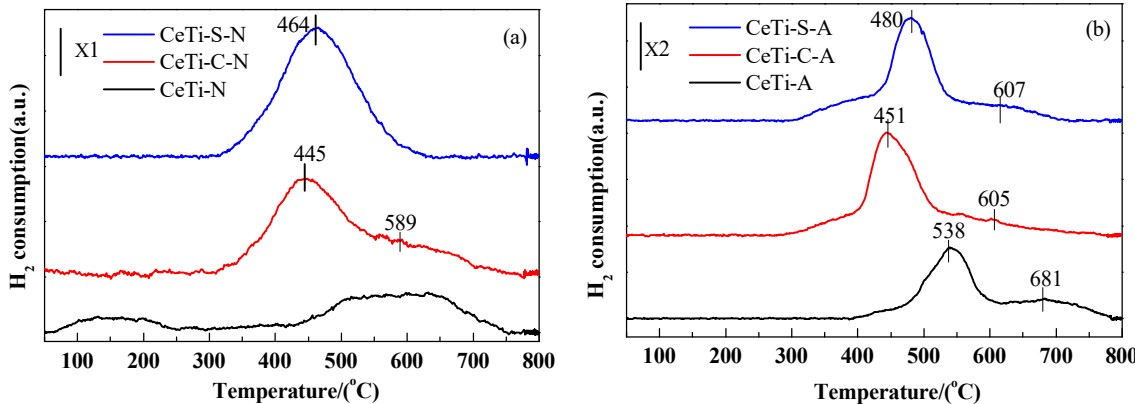

**Figure 6.** H$_2$-TPR profiles of the CeO$_2$/TiO$_2$ catalysts calcined in N$_2$ atmosphere (**a**) and in air atmosphere (**b**) respectively.

The adsorption of NH$_3$ on surface acid sites of catalysts was proven crucial for the NH$_3$-SCR reaction [36]. NH$_3$-TPD data is capable of reflecting the number and strength of acid sites on the surface of the catalysts. Figure 7 shows the NH$_3$-TPD profiles of the catalysts. All catalysts showed two broad peaks between 50 °C to 600 °C, which revealed the co-presence of weak acid sites and strong acid sites. The low temperature peak was attributed to the desorption of physiosorbed NH$_3$ on the weak Bronsted acid sites, and the high temperature peak was ascribed to the desorption of chemisorbed NH$_3$ on the strong Lewis acid sites [37,38]. It could be seen that the peaks of CeTi-C-N and CeTi-S-N were broadened under high temperature, revealing the increase of the number of strong Lewis acid sites comparing to the samples of CeTi-C-A and CeTi-S-A, respectively (Table S2).

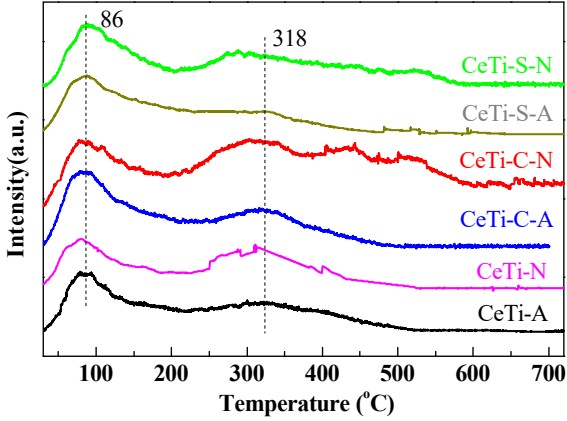

**Figure 7.** NH$_3$-TPD profiles of the CeO$_2$/TiO$_2$ catalysts.

Lewis acid sites were reported to exhibit a higher specific activity than Bronsted acid sites [39]. Interestingly, the amounts of Lewis acid sites on the surface of CeTi-S-A and CeTi-S-N were lower than that on CeTi-C-A and CeTi-C-N, respectively. This might be one reason why CeTi-S-A and CeTi-S-N had bad low temperature activities in the NH$_3$-SCR reaction. In addition, the NH$_3$ desorption peaks of CeTi-C-N and CeTi-S-N were extended to high temperature that improved their low temperature catalytic activity by accelerating the adsorption and activation of NH$_3$. In the meantime, the strong adsorption might lead to excessive oxidation of NH$_3$ to N$_2$O at high temperature. This could be a reason why CeTi-C-N and CeTi-S-N exhibited low activities at a high temperature range.

To explain the great variations of the activities of CeTi-S-A and CeTi-S-N in the presence of vapor and SO$_2$, the SO$_2$-TPD experiments for CeTi-S-A and CeTi-S-N were performed (Figure S5). There are two desorption peaks of SO$_2$ for both of them. For CeTi-S-A, the peaks appeared in the temperature range of 50–200 °C and 630–700 °C, respectively, which might be attributed to the physiosorbed

or weakly adsorbed SO$_2$ and the typical decomposition of cerium sulfate, respectively [40–42]. For CeTi-S-N, a large peak at 50–140 °C and a wide peak at 150–400 °C were observed, indicating that more SO$_2$ molecules adsorbed on the surface of CeTi-S-N than that on CeTi-S-A, which leaded to more active sites being locked. No SO$_2$ desorption peak at high temperature range was detected revealed that almost no cerium sulfate formed on CeTi-S-N surface, possibly due to the inhibition of carbon residue in this sample.

### 2.5. Surface Chemical State Analysis (XPS)

To clarify the surface composition and chemical states of the catalysts, XPS studies were performed. The surface atomic concentrations of Ce, Ti and O were determined using the Peak-Fit program. The XPS spectrum of Ce 3d could be divided into two sets, which could be attributed to 3d$_{3/2}$ and 3d$_{5/2}$ spin-orbital multiplets after fitting. In Figure 8a, the Ce 3d$_{5/2}$ and Ce 3d$_{3/2}$ peaks were titled as v and u, respectively. The bands labeled u$^0$, u′ and v$^0$, v′ represent the 3d$^{10}$4f$^1$ initial electronic state, corresponding to Ce$^{3+}$, while other peaks (u, u″, u‴, v, v″, v‴) are owing to a 3d$^{10}$4f$^0$ state that represent Ce$^{4+}$ species, indicating that Ce$^{3+}$ and Ce$^{4+}$ species coexist over these CeO$_2$/TiO$_2$ catalysts simultaneously [43,44]. The level of Ce$^{3+}$/(Ce$^{3+}$ +Ce$^{4+}$) of the catalysts calcined in N$_2$ atmosphere was shown to be slightly larger than that of the samples calcined in the air. Otherwise, the surface atomic concentrations of the samples were investigated by Peak-Fit the values (Table S3). The results indicated that the introduction of organic assistants in the milling process might have a slight promotion on the surface atomic concentration of Ce on the CeO$_2$/TiO$_2$ catalysts, further indicating a higher dispersion of cerium species. Obviously, the carbon residue on the surface of the catalysts could not be neglected. The surface atomic concentration of carbon on CeTi-S-N was much higher than that of CeTi-C-N, revealing that more active sites were covered by the residue carbon atoms on CeTi-S-N compaed to CeTi-C-N. This phenomenon was also supported by TEM data.

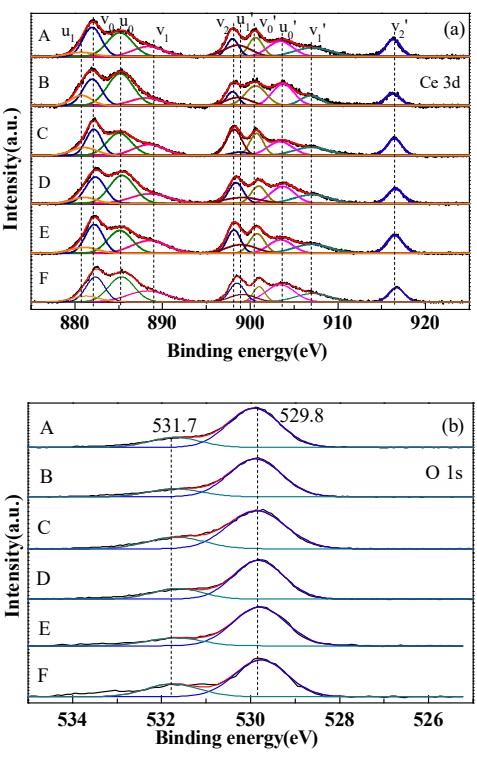

**Figure 8.** *Cont.*

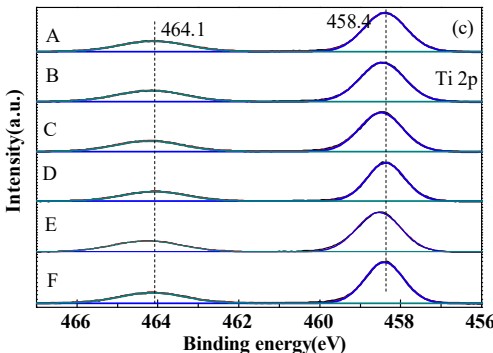

**Figure 8.** XPS patterns of the Ce 3d (**a**) O 1s (**b**) and Ti 2p (**c**) spectrum of various catalysts: A. CeTi-A; B. CeTi-N; C. CeTi-C-A; D. CeTi-C-N; E. CeTi-S-A; F. CeTi-S-N.

The O 1s XPS results of all catalysts are shown in Figure 8b. All peaks were fitted into two groups referred to the lattice oxygen (labeled as $O_{latt}$) at 529.8 eV and the surface chemisorbed oxygen (labeled as $O_{sur}$) at 531.7 eV [45], e.g. a hydroxyl-like groups. The levels of $O_{sur}$ on $CeO_2/TiO_2$ catalysts were shown in order: CeTi-S-N > CeTi-C-N > CeTi-S-A > CeTi-C-A > CeTi-A > CeTi-N, indicating that the introducing of organic assistants during the ball-milling process could improve the $O_{sur}$ ratio of the $CeO_2/TiO_2$ catalysts. But this was not in agreement with the $NH_3$-SCR activities over these catalysts, which implies that the $O_{sur}$ ratio of $CeO_2/TiO_2$ catalyst might not be the factor dominating the catalytic activity in the $NH_3$-SCR reaction. The chemisorbed oxygen that was adsorbed on the defective sites of catalysts exhibited good migration ability, which promoted oxidize NO to $NO_2$ and subsequently improved the catalytic activity for $NH_3$-SCR by the "fast SCR" approach [46]. The Ti 2p spectra results of all catalysts are given in Figure 8c. The peaks at 458.4 eV and 464.1 eV were assigned to Ti $2p_{3/2}$ and Ti $2p_{1/2}$, respectively [25]. There had hardly ever been binding energy shift in all catalysts, indicating that the introduction of organic assistant in the milling process had little effect on the valence states of titanium species.

## 3. Experimental

### 3.1. Preparation of $CeO_2/TiO_2$ Catalysts

In the presence or absence of organic matters (citric acid and sucrose), the $CeO_2/TiO_2$ catalysts were prepared by dry ball-milling using cerium (III) nitrate hexahydrate as the cerium precursor. Titanium precursor, commercial anatase $TiO_2$ (from Fangxinlihua Company, Beijing, China), using 20 wt% ceria in the total $CeO_2/TiO_2$ catalyst, were placed into a 500 $cm^3$ sintered zirconium oxide grinding jar with agate balls (20, 15 and 10 mm in diameter). The ball-to-powder mass ratio was 10:1, and the cerium precursor-to-organic matter mass ratio reached 1:1. Milling was performed in a QM-QX planetary ball mill (Nanjing NanDa Instrument Plant, Nanjing, China) for 1 h at a rotation speed of 500 rpm. The milled mixture was calcined in air or $N_2$ atmosphere for 2 steps, respectively, at 250 °C for 2 h first and then at 500 °C in a ramping rate of 5 °C·$min^{-1}$ for 2 h. The catalyst prepared by dry ball milling with cerium (III) nitrate hexahydrate as the cerium precursor without organic matter calcined in air atmosphere was labelled as "CeTi-A", and the corresponding calcined in $N_2$ was labelled as "CeTi-N". The samples prepared by 1:1 citric acid-assisted ball milling calcined in air and $N_2$ atmosphere are denoted as "CeTi-C-A" and "CeTi-C-N", respectively. Samples labelled as "CeTi-S-A" and "CeTi-S-N" refer to the catalysts prepared with sucrose as the organic matter-assisted ball milling calcined in air and $N_2$, respectively.

### 3.2. $NH_3$-SCR Catalytic Activity Test

The $NH_3$-SCR catalytic activity measurement was performed in a fixed-bed quartz flow reactor (i.d. = 12 mm). The catalyst sample loaded to the reactor was 0.2 g with particle size of 40–60 mesh.

The reaction temperature was monitored using a K-type thermocouple inserted into the catalyst bed. Reaction gases were controlled using mass-flow controllers before entering the reactor. The concentrations of the simulated flue gases included: 1000 ppm NO, 1000 ppm $NH_3$, 6 vol % $O_2$, $H_2O$ 10% (when added), $SO_2$ 200 ppm (when added) and $N_2$ balanced; the gas hourly space velocity (GHSV) was up to 30,000 $h^{-1}$. NO concentrations in the inlet and outlet gas were continually analyzed using a 4000 VM Heated Vacuum $NO_x$ gas analyzer (from Signals group, Camberley, United Kingdom). The products (e.g. NO, $NH_3$, $NO_2$ and $N_2O$) were analyzed using an online FT-IR spectrometer (Bruker Tensor 27 from BRUKER, Karlsruhe, Germany) equipped with a gas cell with 2.4 m optical length. The spectra were collected samples after the SCR reaction had reached a steady state at a specific temperature. The NO conversion and the $N_2$ selectivity were calculated by:

$$\text{NO conversion} = \frac{[\text{NO}]_{in} - [\text{NO}]_{out}}{[\text{NO}]_{in}} \times 100\% \tag{1}$$

$$\text{N}_2 \text{ selectivity} = \frac{[\text{NO}]_{in} + [\text{NH}_3]_{in} - [\text{NO}_2]_{out} - 2[\text{N}_2\text{O}]_{out}}{[\text{NO}]_{in} + [\text{NH}_3]_{in}} \times 100\% \tag{2}$$

*3.3. Characterization of the Catalysts*

The textural properties of the catalyst samples were determined by nitrogen adsorption-desorption at 77 K with an ASAP 2020 Plus 1.03 (from Micromeritics Instrument Corp., Norcross, GA, USA) chemisorption instrument. The specific surface area and the pore size distribution were calculated using the Barrett-Emmett-Teller (BET) method and the Barrett-Joyner-Halenda (BJH) method, respectively. The pore volume and pore volume distribution were determined by the adsorption isotherms. The X-ray diffraction (XRD) patterns were recorded on a Bruker D8 ADVANCE (from BRUKER, Karlsruhe, Germany) diffractometer operating at 40 kV and 40 mA with Cu Kα radiation (λ = 0.15406 nm). The data was collected in the scattering angle range of 20–80°, with a step size of 0.02°. The X-ray photoelectron spectroscopy (XPS) studies were performed on Thermo ESCALAB 250XI equipment (from ThermoFisher Scientific at Waltham, MA, USA) with Al Kα X-ray radiation under UHV. All the binding energy values were calibrated based on the criterion of C1s (B.E. = 284.8 eV). The morphology of the catalysts was characterized under high-resolution transmission electron microscopy (HR-TEM) on a FEI Tecnai G220 apparatus (from FEI at Hillsboro, USA) at 200 kV. Raman spectra were collected using a HORIBA LabRAM HR Evolution Laser Raman spectrometer (from HORIBA Scientific, Kyoto, Japan). The laser of excitation wavelength was 532 nm. $H_2$-temperature programmed reduction ($H_2$-TPR) was performed in a quartz reactor, and about 50 mg of sample was sieved through 40–60 mesh. First, the sieved sample was pre-treated in Ar (30 mL·$min^{-1}$) at 200 °C for 1 h and then cooled to ambient temperature. Next, a flowing mixture of 5% $H_2$/Ar (30 ml·$min^{-1}$) was switched on at a linear heating rate of 10 °C·$min^{-1}$. The $H_2$ consumption signal was continuously monitored by AutoChem II 2920 Chemisorption instrument (from Micromeritics Instrument Corp., Norcross, GA, USA) with a thermal conductivity detector (TCD). $NH_3$ and $SO_2$ temperature programmed desorption (TPD) experiments were performed in a quartz reactor. About 100 mg samples with a particle size of 40–60 mesh were loaded to the reactor. Before the TPD, all samples were pretreated with high-purity (99.999%) $N_2$ (30 ml/min) at 200 °C for 1 h and then saturated with 1.03% ammonia that was balanced by $N_2$ ($NH_3$-TPD) or 0.51 % sulfur dioxide that was balanced by $N_2$ ($SO_2$-TPD) at ambient temperature for 1 h, subsequently swept by $N_2$ at the same temperature for 0.5 h to remove weakly bound (physiosorbed) ammonium or sulfur dioxide. Finally, the TPD operation was performed from 30 to 800 °C at a heating rate of 10 °C·$min^{-1}$. The amount of $NH_3$ or $SO_2$ desorbed was monitored using Tianjin XQ TP-5080 (Tianjin, China) auto-adsorption apparatus with a TCD and Hiden QGA mass spectrum analyzer, respectively. Thermogravimetric analysis (TG) was carried out on a Netzsch thermal analyzer (STA449C) (New South Wales, Australia) with a heating rate of 10 °C /min in a flowing air or nitrogen with rate of 100 mL/min. The samples of mixtures of $Ce(NO_3)_3$/$TiO_2$/citric-acid, $Ce(NO_3)_3$/$TiO_2$/sucrose

and Ce(NO$_3$)$_3$/TiO$_2$ were prepared by grinding using Ce(NO$_3$)$_3$·6H$_2$O (0.253 g), TiO$_2$(0.4 g) and citric acid (C$_6$H$_8$O$_7$·H$_2$O, 0.253 g), or Ce(NO$_3$)$_3$·6H$_2$O (0.253 g), TiO$_2$(0.4 g) and sucrose (C$_{12}$H$_{22}$O$_{11}$, 0.253 g), or Ce(NO$_3$)$_3$·6H$_2$O (0.253 g) and TiO$_2$(0.4 g), respectively. About 5–8 mg of mixture was used for the TG test. FT-IR spectra were performed on a Bruker Tensor II instrument (KBr pellets), which were collected at atmospheric pressure and room temperature with a resolution of 4 cm$^{-1}$ and accumulation of 32 scans. The samples for FT-IR were also prepared by grinding firstly using the same precursors as that of TG, and then were treated under the desired temperature for 2 h in air.

## 4. Conclusions

To sum up, six 20 wt% CeO$_2$/TiO$_2$ catalysts were prepared by dry ball milling in the absence and presence of organic assistants. It was found that the presence of citric acid could change the decomposition process of the precursor mixture significantly, which could in turn lead to improvement of the dispersion and reducibility of cerium species, the surface acidity as well as the surface microstructure, all which accounted for the high low temperature activity of CeTi-C-N in the NH$_3$-SCR reaction. However, the addition of sucrose during the ball milling had little effect on the decomposition process of the corresponding precursor mixture. The activity data of CeTi-S-N and CeTi-S-A revealed that adding sucrose seemed to have an inhibitory effect on the catalytic performance of CeO$_2$/TiO$_2$ catalysts in NH$_3$-SCR reaction, possibly due to the decrease in the crystallinity of the TiO$_2$ support and the carbon residue covering the part active sites on the catalyst surface. In addition, calcining treatment in nitrogen atmosphere promoted the formation of the surface defects and Lewis acid sites on the catalysts, which improved the low temperature catalytic activity while decreasing the N$_2$ selectivity due to the promotion of excessive oxidation of NH$_3$ to N$_2$O at high temperature.

**Supplementary Materials:** The following are available online at http://www.mdpi.com/2073-4344/9/4/357/s1, Figure S1: Catalytic activities of the CeO$_2$/TiO$_2$ catalysts in the presence of SO$_2$ and H$_2$O, Figure S2: Nitrogen adsorption-desorption isotherms, Figure S3: TG profiles of the mixtures of Ce(NO$_3$)$_3$/TiO$_2$, Figure S4: FT-IR results, Figure S5: SO$_2$-TPD profiles. Table S1: H$_2$ consumption amounts, Table S2: NH$_3$-TPD data, Table S3: Data of XPS.

**Author Contributions:** X. H. and S. L. designed and carried out the experiments and wrote part of the manuscript; J. C. did part of the data treatment; Y. C. and Y. S. did part of the samples characterization; G. B. and L. S. designed part of the experiments; G. Z. did part of the data analyzing and project administration; H. H. was the supervision and edited the manuscript; W. Q. was the supervision and wrote and edited the manuscript.

**Acknowledgments:** The work was supported by the National Natural Science Foundation of China (201577005) and the National Key R&D Program of China (2017YFC0210303-02).

**Conflicts of Interest:** The authors declare no conflict of interest.

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
