# Peer review of "Effect of Organic Assistant on the Performance of Ceria-Based Catalysts for the Selective Catalytic Reduction of NO with Ammonia"

_catalysts, doi:10.3390/catal9040357_

Round 1

Reviewer 1 Report

In this paper the authors investigated the effect of the addition of two organics (citric acid and sucrose) in the preparation of CeO2/TiO2 catalysts by ball-milling starting from cerium (III) nitrate and commercial TiO2.

The advantage of using this method compared to more traditional wet or incipient wetness impregnation techniques, starting from aqueous solution of cerium nitrate, is not clear in this manuscript and must be explained as done in ref. [21]. Anyway, a method involving the introduction of an organic compound instead of distilled water is quite complex and scarcely controllable, including possible additional reactions between the organic compound and nitrates during the calcination which, on the contrary, are limited to nitrates decomposition when an aqueous solution is used.

Properties of TiO2 used must be reported. The support is supposedly the same used in ref. [21] having 77 m2/g BET surface area. As a consequence, on the basis of data reported in Table 1, it seems that the absence of any organic assistant in the ball milling reduces the original surface area of the support (both calcining under air or N2), the presence of citric acid does not change this value (both calcining under air or N2) whereas the presence of sucrose increases the BET area only when nitrogen is used in the thermal treatment. All these results suggest that the choice of the organic assistant and of the gas used in the calcination step is important because it affects the reactions occurring during the transformation of cerium nitrate into cerium oxide. Nevertheless, although the authors deeply characterize the catalysts trying, with little success, to correlate the performance in SCR reaction with some properties, they totally neglect to investigate the most important step determining the final properties of the catalysts through techniques such as TG analysis, MS analysis etc.

Indeed, during the thermal treatment, in addition to the decomposition of nitrates, reactions between evolved NOx and organics can occur in the absence of O2, similar to what happens in a TWC catalyst between NOx and unburned hydrocarbons. On the other hand, combustion of organics, with possible hotspots, can occur under air. Both treatments seem not independent of the structure of the organic compound. Unfortunately, the authors do not investigate this aspect.

Finally, the presence of carbonaceous residues on the catalyst is not acceptable for a good preparation technique.

Conclusions are totally arbitrary. CeTi-C-N catalyst (the best performing one) is not the best reducible, or that with the highest amount of surface oxygen or with the highest surface area, as the authors claim. It just has a larger amount of strong acid sites which could be responsible for the better performance.

To sum up, the authors should add a study of phenomena occurring during the thermal treatments of milled catalysts in order to understand how the organic assistant interacts with nitrates and TiO2.

Other questions:

How the ball milling with or without the organic assistant affect the TiO2? According to the authors, does the preparation method they used reduce the size of CeO2 crystallites or of TiO2 (or of both)?

What is the size of catalyst particles they used in the SCR reactor? Most likely they did not use the powder after 1h milling as it is, due to the high pressure drop it can cause in the reactor.

In Fig. S1 the hysteresis of CeTi-S-N is significantly shifted towards low p/p0 values. How does that affect the pore size distribution and, in particular, the mesopores volume? What does it mean a so small average pore diameter (Table 1)? What kind of pores are occluded with carbon residues? Why this catalyst almost totally loose the crystalline structure of TiO2 (Fig. 2-XRD)? (please, use always the same colour to identify each catalyst. Changing colours is very confusing).

The only information deducible from TPR analysis is a higher CeO2 dispersion in the catalysts prepared with the organic assistant since H2 uptake was related to surface oxygen and not to reduction of Ce(IV) to Ce(III).

No correlation with catalytic performance can be found.

What is the low temperature (100-200°C) peak of CeTi-N catalyst in Fig. 5a?

Author Response

Answers to Comments by Referees

Reviewer 1

Comments and Suggestions for Authors:

In this paper the authors investigated the effect of the addition of two organics (citric acid and sucrose) in the preparation of CeO2/TiO2 catalysts by ball-milling starting from cerium (III) nitrate and commercial TiO2.

The advantage of using this method compared to more traditional wet or incipient wetness impregnation techniques, starting from aqueous solution of cerium nitrate, is not clear in this manuscript and must be explained as done in ref. [21]. Anyway, a method involving the introduction of an organic compound instead of distilled water is quite complex and scarcely controllable, including possible additional reactions between the organic compound and nitrates during the calcination which, on the contrary, are limited to nitrates decomposition when an aqueous solution is used.

Answer: We have revised the introduction of the manuscript according to the reviewer’s suggestion.

Properties of TiO2 used must be reported. The support is supposedly the same used in ref. [21] having 77 m2/g BET surface area. As a consequence, on the basis of data reported in Table 1, it seems that the absence of any organic assistant in the ball milling reduces the original surface area of the support (both calcining under air or N2), the presence of citric acid does not change this value (both calcining under air or N2) whereas the presence of sucrose increases the BET area only when nitrogen is used in the thermal treatment. All these results suggest that the choice of the organic assistant and of the gas used in the calcination step is important because it affects the reactions occurring during the transformation of cerium nitrate into cerium oxide. Nevertheless, although the authors deeply characterize the catalysts trying, with little success, to correlate the performance in SCR reaction with some properties, they totally neglect to investigate the most important step determining the final properties of the catalysts through techniques such as TG analysis, MS analysis etc.

Answer: We have added the textural and structural data of TiO2 support in Table 1, and revised the statement in corresponding paragragh. In addition, the TG and FT-IR analysis were also supplied.

The surface areas of CeTi-A and CeTi-N were even lower than the TiO2 support possibly due to the addition of ceria that blocked the partial pores of TiO2.

Indeed, during the thermal treatment, in addition to the decomposition of nitrates, reactions between evolved NOx and organics can occur in the absence of O2, similar to what happens in a TWC catalyst between NOx and unburned hydrocarbons. On the other hand, combustion of organics, with possible hotspots, can occur under air. Both treatments seem not independent of the structure of the organic compound. Unfortunately, the authors do not investigate this aspect.

Finally, the presence of carbonaceous residues on the catalyst is not acceptable for a good preparation technique.

Answer: We have supplied the TG and FT-IR data in the manuscript that might give an interpretation about the above question. But the reactions between NOx and organics are too complicated to explain in the present state.

It is true that the presence of carbonaceous residues on the catalyst is not acceptable for a catalyst preparation technique. Here we just want to discuss the effects of the decomposition of organic assistant at different atmosphere. Further research is still in the works.

Conclusions are totally arbitrary. CeTi-C-N catalyst (the best performing one) is not the best reducible, or that with the highest amount of surface oxygen or with the highest surface area, as the authors claim. It just has a larger amount of strong acid sites which could be responsible for the better performance.

To sum up, the authors should add a study of phenomena occurring during the thermal treatments of milled catalysts in order to understand how the organic assistant interacts with nitrates and TiO2.

Answer: We have revised the conclusion and the related part according to the reviewer’s suggestion.

In the CeTi-S-N sample, the crystallinity of TiO2 decreased significantly, but it is still not easy to understand the interaction between organic assistant, nitrates and TiO2. The change of the crystallinity of TiO2 possibly attributes to the presence of carbon residue.

Other questions:

How the ball milling with or without the organic assistant affect the TiO2? According to the authors, does the preparation method they used reduce the size of CeO2 crystallites or of TiO2 (or of both)?

Answer: We can ascertain that the ball milling in the presence of organic assistant can reduce the size of CeO2 crystallites due to the interaction of organic assistant and cerium ions. But the effect on the TiO2 was not very clear.

What is the size of catalyst particles they used in the SCR reactor? Most likely they did not use the powder after 1h milling as it is, due to the high pressure drop it can cause in the reactor.

Answer: The particle size of the catalysts used in the catalytic activity test was about 40-60 mesh. We have revised the statement in the experimental.

In Fig. S1 the hysteresis of CeTi-S-N is significantly shifted towards low p/p0 values. How does that affect the pore size distribution and, in particular, the mesopores volume? What does it mean a so small average pore diameter (Table 1)? What kind of pores are occluded with carbon residues? Why this catalyst almost totally loose the crystalline structure of TiO2 (Fig. 2-XRD)? (please, use always the same colour to identify each catalyst. Changing colours is very confusing).

Answer: We have revised the statement in section 3.2. We also change the line colour in Fig. 2.

The only information deducible from TPR analysis is a higher CeO2 dispersion in the catalysts prepared with the organic assistant since H2 uptake was related to surface oxygen and not to reduction of Ce(IV) to Ce(III).

No correlation with catalytic performance can be found.

What is the low temperature (100-200°C) peak of CeTi-N catalyst in Fig. 5a?

Answer: In the reduction atmosphere, Ce(IV) will be reduced to Ce(III), so the hydrogen consumption attributes to the reduction of surface absorbed oxygen, surface lattice oxygen and bulk oxygen. The reduction of surface lattice oxygen and bulk oxygen is agree with the reduction of surface and bulk Ce(IV) ions.

The low temperature peak (100-200°C) of CeTi-N can be assigned to the reduction of surface absorbed oxygen. Because heat-treating CeO2 under nitrogen atmosphere may lead to the formation more oxygen vacancy.

Reviewer 2 Report

The manuscript is focused on the interesting topic to study the effect of organic dopant on the performance of ceria-based catalysts used in the selective catalytic reduction of NO to ammonia. Based on the title and the abstract of the manuscript I was very keen to do the review of this contribution to be published.

The manuscript is very well constructed and a range of characterisation techniques has been applied to understand the promoting/inhibiting effect on the catalytic materials for SCR. The data interpretation is done carefully and findings are cross-correlated between different techniques. However, I believe the manuscript could be improve by focusing on the following points:

            ·         Line 89 - sample preparation for each technique should be included

·         Figure 1c – it should be specify, otherwise it looks like Fig1a is under the same conditions as Fig1c

·         Table 1 – what is the error bar of the obtained values?

·         Figure 3a – it is interesting that no signal for crystalline CeO2 is present in Raman data as the crystalline form is clearly seen in XRD and TEM. Thus, the explanation that there is no crystallinity for CeO2 does not apply. Please expand the discussion on that.

·         Figure 3b – there is quite significant amount of C produced for CeTi-C-N (slightly smaller than for CeTi-S-N, however it shows it is the best low-T SCR catalysts. The discussion should be expanded why this is the case. Otherwise, the explanation for the less active catalysts (CeTi-S-N) due to the C formation does not apply directly.

Can the Raman signal be quantified and compared to the XPS data presented later in the manuscript?

·         Line 217 – it should be Fig 4c

·         Line 238 – it should be Fig 5b

·         Line 243 – it should be Fig 5a

·         Line 254 – at this point in the manuscript the XPS data are not introduced, thus the comparison should not be mentioned. Otherwise, it will be confusing for the reader.

·         Table S3 – the errors should be included in the calculated data.

·         C 1s XPS quantification  should be compared to the Raman C signal

·         I do not agree with the fitting of Ce 3d XPS. There is a number of classic papers on Ce 3d XPS, where 10 overall signals are corresponding to the Ce3+/Ce4+ spectrum. The relevant papers are: Larachi et al. Appl. Surf. Sci., 2002, 195, 235-250, Mekki et al. J. Electron. Spectrosc. Relat. Phenom., 2005, 142, 75-81, Beche et al, Surf. Interface Anal. 2008; 40: 264–267

Author Response

Answers to Comments by Referees Reviewer 2 Comments and Suggestions for Authors The manuscript is focused on the interesting topic to study the effect of organic dopant on the performance of ceria-based catalysts used in the selective catalytic reduction of NO to ammonia. Based on the title and the abstract of the manuscript I was very keen to do the review of this contribution to be published. The manuscript is very well constructed and a range of characterisation techniques has been applied to understand the promoting/inhibiting effect on the catalytic materials for SCR. The data interpretation is done carefully and findings are cross-correlated between different techniques. However, I believe the manuscript could be improve by focusing on the following points: •Line 89 - sample preparation for each technique should be included Answer: We have added the states of catalyst samples in the experimental part. •Figure 1c – it should be specify, otherwise it looks like Fig1a is under the same conditions as Fig1c. Answer: For clear, we have removed the Fig 1c in the manuscript to the support information. • Table 1 – what is the error bar of the obtained values? Answer: Usually the relative error of the BET test is several per cent. So the total pore volume of CeTi-S-N (0.13) should be similar to the data of CeTi-A and CeTi-N (0.11). We have revised the description in the manuscript. • Figure 3a – it is interesting that no signal for crystalline CeO2 is present in Raman data as the crystalline form is clearly seen in XRD and TEM. Thus, the explanation that there is no crystallinity for CeO2 does not apply. Please expand the discussion on that. Answer: We have revised the statement in the corresponding paragraph according to the reviewer’s suggestion. • Figure 3b – there is quite significant amount of C produced for CeTi-C-N (slightly smaller than for CeTi-S-N, however it shows it is the best low-T SCR catalysts. The discussion should be expanded why this is the case. Otherwise, the explanation for the less active catalysts (CeTi-S-N) due to the C formation does not apply directly. Answer: Although the Raman signals of carbon species in CeTi-C-N and CeTi-S-N are similar, the particle morphology and existing form of the carbon residue are different. We have discussed this difference in line 238-243. Can the Raman signal be quantified and compared to the XPS data presented later in the manuscript? Answer: It is difficult to quantify and compare the data of XPS and Raman spectra. •Line 217 – it should be Fig 4c •Line 238 – it should be Fig 5b •Line 243 – it should be Fig 5a Answer: We have revised these rookie mistakes in the manuscript according to the reviewer’s suggestion. •Line 254 – at this point in the manuscript the XPS data are not introduced, thus the comparison should not be mentioned. Otherwise, it will be confusing for the reader. Answer: We have revised the manuscript according to the reviewer’s suggestion. •Table S3 – the errors should be included in the calculated data. Answer: We know that the relative error of the XPS test is also in several per cent level. So we have revised the statement in the corresponding paragraph. •C 1s XPS quantification should be compared to the Raman C signal Answer: It is difficult to quantify and compare the data of XPS and Raman spectra. The surface atomic concentration of C of CeTi-N, CeTi-C-N and CeTi-S-N were received by subtracting the C signal of CeTi-A, CeTi-C-A and CeTi-S-A from the C signal of CeTi-N, CeTi-C-N and CeTi-S-N, respectively. The results can show the difference of surface C atomic concentration of CeTi-N, CeTi-C-N and CeTi-S-N. •I do not agree with the fitting of Ce 3d XPS. There is a number of classic papers on Ce 3d XPS, where 10 overall signals are corresponding to the Ce3+/Ce4+ spectrum. The relevant papers are: Larachi et al. Appl. Surf. Sci., 2002, 195, 235-250, Mekki et al. J. Electron. Spectrosc. Relat. Phenom., 2005, 142, 75-81, Beche et al, Surf. Interface Anal. 2008; 40: 264–267 Answer: We have revised the XPS data and Figure in the manuscript according to the reviewer’s suggestion.

Reviewer 3 Report

The authors presented a study of the influence of organic assistant (citric acid and sucrose) on the performance of ceria-based catalysts. The paper is good with interesting findings. But there are some minor issues to address:

1.      It should be noted somewhere in the text why authors pick the citric acid and sucrose for testing?

2.      Page 3 line 108-109 sentence should be clarified.   

3.      Page 3 line 109 ”should be pretreated” need to be corrected to ”were pretreated”. The ”should” imply that samples might or might not be pretreated.

4.      Figure 1 subfigure c) it is not clear from Figure and from the label that results on subfigure c) is when H2O and SO2 are present.

5.      Page 12 References should be added to XPS data for O2.

Author Response

Answers to Comments by Referees Reviewer 3 Comments and Suggestions for Authors The authors presented a study of the influence of organic assistant (citric acid and sucrose) on the performance of ceria-based catalysts. The paper is good with interesting findings. But there are some minor issues to address: 1. It should be noted somewhere in the text why authors pick the citric acid and sucrose for testing? Answer: We have supplied the corresponding statement in the introduction according to the reviewer’s suggestion. 2. Page 3 line 108-109 sentence should be clarified. Answer: We have revised the corresponding sentences according to the reviewer’s suggestion. 3. Page 3 line 109 ”should be pretreated” need to be corrected to ”were pretreated”. The ”should” imply that samples might or might not be pretreated. Answer: We have revised the corresponding sentences according to the reviewer’s suggestion. 4. Figure 1 subfigure c) it is not clear from Figure and from the label that results on subfigure c) is when H2O and SO2 are present. Answer: For clear, we have removed the Fig 1c in the manuscript to the support information. 5. Page 12 References should be added to XPS data for O2. Answer: We have supplied the reference according to the reviewer’s suggestion.

Round 2

Reviewer 1 Report

The authors answered to most of questions and also included additional TG and FTIR experiments which provided new insights. Nevertheless, there are still some questions:

TG

Line 254-255. How do the authors estimate the theoretical final mass loss? Why is it different if operating under N2 or air (64.7% and 55.2% respectively)? Don’t they hypothesize that all organic assistant and nitrates completely decompose (or are burnt in the case of the organic assistant under air) leaving CeO2 and TiO2?

The different final weight achieved for samples prepared with both organic assistant suggests that treatment under N2 always leave organic residues.

Comparison with a blank run of ball-milled Ce(NO3)3/TiO2 (when only nitrates decomposition is expected to take place) could provide more information about the possible interaction between nitrates and the organic assistant.

Fig.5. The main difference between TG curves of catalysts with citric acid and sucrose is that in the case (a) treatments under N2 and air follow the same trend even if at T>200°C the N2 curve is always at higher weight levels, whereas in the case (b) the N2 curve does not have a well detectable step 3. Please, explain the absence of this step.

FTIR

Fig. S3. How has the FTIR analysis been done? Under controlled atmosphere? How has the catalyst been treated? Under air or N2? In-situ or ex-situ?

Fig. S3b shows in the fingerprint region (at 20200px-1) the crystal Ti-O-Ti vibrations of TiO2 whereas this band is absent in Fig. S3a. How do the authors explain that?

TPR

Authors answer: In the reduction atmosphere, Ce(IV) will be reduced to Ce(III), so the hydrogen consumption attributes to the reduction of surface absorbed oxygen, surface lattice oxygen and bulk oxygen. The reduction of surface lattice oxygen and bulk oxygen is agree with the reduction of surface and bulk Ce(IV) ions.

Reduction of CeO2 at 450-550°C is assigned to reduction of surface ceria. The lower reduction temperature of this reduction for the catalysts prepared using the organic assistant is a clear indication of a higher CeO2 dispersion.

Details about TG and FTIR experiments must be included in the Experimental.

Fourier (Capital F) in the abstract.

Author Response

Answers to Comments by Referees

Reviewer 1

Review Report Form

Open Review

English language and style

( ) Extensive editing of English language and style required
(x) Moderate English changes required
( ) English language and style are fine/minor spell check required
( ) I don't feel qualified to judge about the English language and style

Yes

Can be improved

Must be improved

Not applicable

Does the introduction   provide sufficient background and include all relevant references?

(x)

( )

( )

( )

Is the research design   appropriate?

( )

(x)

( )

( )

Are the methods adequately   described?

( )

(x)

( )

( )

Are the results clearly   presented?

( )

(x)

( )

( )

Are the conclusions   supported by the results?

( )

(x)

( )

( )

Comments and Suggestions for Authors

The authors answered to most of questions and also included additional TG and FTIR experiments which provided new insights. Nevertheless, there are still some questions:

TG

Line 254-255. How do the authors estimate the theoretical final mass loss? Why is it different if operating under N2 or air (64.7% and 55.2% respectively)? Don’t they hypothesize that all organic assistant and nitrates completely decompose (or are burnt in the case of the organic assistant under air) leaving CeO2 and TiO2?

Answer: The calculated mass loss was estimated according to the precursor’s molecular formula hypothesizing that the crystal water, nitrate compound, organic assistant, and etc decomposed completely under the corresponding temperature ranges. The difference of the residual mass fractions in N2 condition and that in air condition is resulted from the existence of carbon residue.

 The different final weight achieved for samples prepared with both organic assistant suggests that treatment under N2 always leave organic residues.

Answer: Yes, treatment under N2 atmosphere leaves residue in the final samples. After calcined at 250 oC and 500 oC, respectively, the organic assistant should have been carbonized almost fully, so the residue was carbon-based species. The different final weight indicated the different carbon content of the organic assistant.

Comparison with a blank run of ball-milled Ce(NO3)3/TiO2 (when only nitrates decomposition is expected to take place) could provide more information about the possible interaction between nitrates and the organic assistant.

Answer: This is a good suggestion. We have add the TG data of the mixture of Ce(NO3)3/TiO2 in the support information and also revised the description in the text.

Fig.5. The main difference between TG curves of catalysts with citric acid and sucrose is that in the case (a) treatments under N2 and air follow the same trend even if at T>200°C the N2 curve is always at higher weight levels, whereas in the case (b) the N2 curve does not have a well detectable step 3. Please, explain the absence of this step.

Answer: In Fig.5b, the N2 curve has a relative short mass loss due to the formation of carbon residue. But in the air condition, almost all organic assistant decomposed up to 500 oC, so the mass loss in this step was detectable.

FTIR

Fig. S3. How has the FTIR analysis been done? Under controlled atmosphere? How has the catalyst been treated? Under air or N2? In-situ or ex-situ?

Answer: We have supplied the experimental conditions of FT-IR in the experimental section. All the samples were treated under desired temperature in air, and the FT-IR spectra collection was ex-situ.

Fig. S3b shows in the fingerprint region (at 20200px-1) the crystal Ti-O-Ti vibrations of TiO2 whereas this band is absent in Fig. S3a. How do the authors explain that?

Answer: The significant difference in the low wavenumber range in Fig.S3a and Fig.S3b was possible due to the equipment problem, because the spectra in Fig.S3a and Fig.S3b were collected on two different equipments. Now we have given the data collected on the same equipment as used in Fig.S3b.

TPR

Authors answer: In the reduction atmosphere, Ce(IV) will be reduced to Ce(III), so the hydrogen consumption attributes to the reduction of surface absorbed oxygen, surface lattice oxygen and bulk oxygen. The reduction of surface lattice oxygen and bulk oxygen is agree with the reduction of surface and bulk Ce(IV) ions.

Reduction of CeO2 at 450-550°C is assigned to reduction of surface ceria. The lower reduction temperature of this reduction for the catalysts prepared using the organic assistant is a clear indication of a higher CeO2 dispersion.

Answer: We have revised the description in the text according to the reviewer’s suggestion.

Details about TG and FTIR experiments must be included in the Experimental.

Answer: We have supplied the experimental conditions of FT-IR and TG.

Fourier (Capital F) in the abstract.

Answer: We have revised the abstract according to the reviewer’s suggestion.

Reviewer 2 Report

The authors responded well to the reviewers comments and that improved the manuscipt significantly. Below, there are just a couple of suggestions: 

Figure 1 - c it should be removed from the caption.

The Ce 3d XPS data - the fitting using overall 10 signals corresponding to Ce3+ and Ce4+ has been applied as suggested. However, more restrictions should be put in place for this analysis. As currently the specific peaks change their positions and band width quite significantly. As a consequence, as the assigned peaks position is very different for all the samples studies, that provides not truthful values to the Ce3+/Ce4+ quantification. The analysis needs to be redone before accepting the manuscript. 

Author Response

Answers to Comments by Referees

Reviewer 2

Review Report Form

Open Review

English language and style

( ) Extensive editing of English language and style required
(x) Moderate English changes required
( ) English language and style are fine/minor spell check required
( ) I don't feel qualified to judge about the English language and style

Yes

Can be improved

Must be improved

Not applicable

Does the introduction   provide sufficient background and include all relevant references?

(x)

( )

( )

( )

Is the research design appropriate?

(x)

( )

( )

( )

Are the methods adequately   described?

( )

(x)

( )

( )

Are the results clearly   presented?

( )

(x)

( )

( )

Are the conclusions   supported by the results?

(x)

( )

( )

( )

Comments and Suggestions for Authors

The authors responded well to the reviewers comments and that improved the manuscipt significantly. Below, there are just a couple of suggestions: 

Figure 1 - c it should be removed from the caption.

Answer: We have revised the text according to the reviewer’s suggestion.

The Ce 3d XPS data - the fitting using overall 10 signals corresponding to Ce3+ and Ce4+ has been applied as suggested. However, more restrictions should be put in place for this analysis. As currently the specific peaks change their positions and band width quite significantly. As a consequence, as the assigned peaks position is very different for all the samples studies, that provides not truthful values to the Ce3+/Ce4+ quantification. The analysis needs to be redone before accepting the manuscript. 

Answer: We have revised the Ce 3d XPS figure (Fig.8a) and data (Table S3) according to the reviewer’s suggestion.

Round 3

Reviewer 2 Report

The manuscipt looks very good and the  suggested changes were applied thus I am pleased to recommend this manuscript for publishing.